# Schwann Cells in Digestive System Disorders

**DOI:** 10.3390/cells11050832

**Published:** 2022-02-28

**Authors:** Karina Goluba, Liga Kunrade, Una Riekstina, Vadims Parfejevs

**Affiliations:** Faculty of Medicine, University of Latvia, House of Science, Jelgavas Str. 3, LV-1004 Riga, Latvia; karina.goluba@lu.lv (K.G.); liga.kunrade@lu.lv (L.K.); una.riekstina@lu.lv (U.R.)

**Keywords:** Schwann cells, digestive system, pancreas, innervation, cancer, neural crest

## Abstract

Proper functioning of the digestive system is ensured by coordinated action of the central and peripheral nervous systems (PNS). Peripheral innervation of the digestive system can be viewed as intrinsic and extrinsic. The intrinsic portion is mainly composed of the neurons and glia of the enteric nervous system (ENS), while the extrinsic part is formed by sympathetic, parasympathetic, and sensory branches of the PNS. Glial cells are a crucial component of digestive tract innervation, and a great deal of research evidence highlights the important status of ENS glia in health and disease. In this review, we shift the focus a bit and discuss the functions of Schwann cells (SCs), the glial cells of the extrinsic innervation of the digestive system. For more context, we also provide information on the basic findings regarding the function of innervation in disorders of the digestive organs. We find diverse SC roles described particularly in the mouth, the pancreas, and the intestine. We note that most of the scientific evidence concerns the involvement of SCs in cancer progression and pain, but some research identifies stem cell functions and potential for regenerative medicine.

## 1. Introduction

The chief functions of our digestive system are to break down food, absorb nutrients, and eliminate waste products. It consists of an alimentary tract, which spans from the mouth through the esophagus, stomach, and bowel, as well as accessory organs, notably the pancreas, liver, and gall bladder [1]. The peripheral nervous system (PNS) exerts control over the digestive system and, depending on the location of neuronal bodies, can be viewed as extrinsic or intrinsic [2]. The intrinsic portion of innervation consists of the enteric nervous system (ENS) and some organ-specific local ganglia [3,4]. 

ENS is the key for proper motility, secretion, and defence in the digestive system and, to some extent, can function independently of the central nervous system (CNS) [5]. Extrinsic innervation, on the other hand, is represented by sympathetic, parasympathetic and sensory branches that are formed mainly by the vagal, spinal, and pelvic nerves (Figure 1) [2]. The functions of extrinsic innervation are in many ways similar to those of ENS, especially in the upper gastrointestinal tract and the accessory organs of the digestive system, but they also help integrate ENS with the CNS [6]. Particularly important are the parasympathetic and sympathetic branches that cooperate to control motility, secretion, and sphincter functions and balance the digestion process [6]. Glial cells are a crucial component of innervation and in recent years much evidence has been accumulated on the important status of ENS glia in the digestive system [7]. In this review, we want to shift the focus and, in particular, discuss the functions of Schwann cells (SC), the glial cells of the extrinsic innervation of the digestive system. Additionally, to give more context, we provide information about organ innervation and summarize what is known about the role of nerves in disease.

## 2. Origins of the PNS, Glial Cell Types and Their Functions

Most of the cells that make up the PNS, such as neurons and Schwann cells (SC), originate from the multipotent [8] neural crest (NC) cells. In fact, all glial cells of the PNS are of NC origin [9,10], while peripheral neurons stem from neurogenic placodes (non-NC origin), migrating NC cells, or later from Schwann cell precursors (SCP) [11,12]. Similarly, most of the major PNS nerves that innervate the digestive system are of NC origin, except for some cranial nerves (e.g., trigeminal and vagal nerves), which contain neurons of both placodal and NC origin [11]. It is proposed that the formation of NC-derived PNS neurons occurs in two waves. Initially, the short-lived multipotent migrating NC cells directly give rise to sensory and sympathetic neurons that have ganglia close to the spinal cord, whereas parasympathetic neurons, which reside in the distant ganglia closer to the target organs, arise from the SCPs migrating along the established preganglionic nerves [12]. Interestingly, the formation of the sacral autonomic outflow, which is generally considered parasympathetic, does not fit this model, since neurons of the pelvic ganglia can form even in the absence of the preganglionic neurons [13]. Furthermore, based on molecular characteristics, sacral autonomic neurons are distinct from those of cranial parasympathetic outflow and instead cluster with sympathetic spinal neurons. This shows how studies of the origins and molecular identity of PNS neurons can challenge the established model to open a discussion on whether the anatomical classification should be revised [14,15,16].

The term SCP refers to a set of cell populations that form directly from NC cells or NC-derived boundary cap cells [17]. SCPs are distinct from the migrating NC cells and have a strong association with the axons. Apart from forming all SCs and many PNS neurons, as mentioned above, they can give rise to other diverse cell phenotypes. Nerve fibroblasts [18], melanocytes [19,20], chromaffin cells [21], dental [22], bone marrow [23], and skeletal mesenchymal cells [24] are all within a broad SCP differentiation range. Together, these studies performed in murine and fish models suggest that SCPs as a common entity are an omnipresent multipotent successor to the NC. However, SCPs in different anatomical locations are heterogeneous [25]. To our knowledge, the exact dynamics of their fate restriction is not fully studied, which spurs a debate on whether some of these cells should indeed be called SCPs [25,26]. SCPs are active during embryonic development and the early postnatal period [27,28]. Whether in mammals these cells persist late in adulthood as nerve-associated progenitors is rather unlikely. It is conceivable that, instead, mature SCs in a situation of physiological stress or injury adopt a more plastic state and contribute to the repair process in various ways, including fate change to produce mesenchymal cells, melanocytes, and neurons [29]. 

The formation of mature SCs from SCPs includes an intermediate stage with a characteristic immature Schwann cell (iSC) phenotype. The iSCs participate in axon sorting and promote nerve maturation. Some iSCs become associated with larger axons in a 1:1 ratio and mostly become myelinating SCs, while others remain associated with several smaller axons and remain non-myelinating [30]. In the organs of the digestive system, the majority of SCs would present either of these two phenotypes, with, for example, non-myelinating SCs being more prevalent in the vagus nerve [31].

It is important to stress that, united by NC origin, PNS glia are not uniform and present a broad spectrum of cells with unique morphology and distinct phenotypes. These differences become apparent already during embryonic development [32]. In addition to myelinating and non-myelinating SCs, satellite glia support neuronal bodies in the ganglia [33]. Less frequent specialized glial cells exist at the tips of the axons, for example, terminal Schwann cells at the neuromuscular junction [34] and SCs forming the sensory end-organs in the skin [35]. Several types of enteric glia populate the digestive tract in massive numbers [36,37] and play key roles in digestive system physiology [7,38].

Perhaps not surprisingly, this rich variety of PNS glial cells fulfil a wide repertoire of functions. In this review we focus on the functions of SCs, some of which might appear novel or non-canonical in the sense that they are not merely restricted to PNS homeostasis [29,39,40]. Among other functions, SCs participate in the maintenance of stem cell niche in the hematopoietic system [41] and pain perception [35,42], and perform stem cell functions in juvenile [27,28] and adult mammals [22]. 

Some of the SC functions become especially apparent in disease settings and injury. In this case, SCs switch to an activated state, meaning they become less differentiated, and start dividing and overexpressing some markers characteristic to glial precursors, for example, nerotrophin receptor p75. Wallerian degeneration that occurs after nerve injury is a classic example of glial cell plasticity, where myelinating SCs change their phenotype dramatically and become specialized repair cells [43]. Similarly, SCs associated with smaller nerve fibers, for example, found in the skin, also react to injury [44,45] and acquire some features of SCPs. In injury and regeneration settings, activated SCs can give rise to new cell types [20,22] or modify the microenvironment in a paracrine way [44,46,47]. 

Tumours are often hyperinnervated and neural signalling contributes to the dynamic tumour environment in many ways [48,49]. Lately, SCs have been functionally implicated in many tumours, including pancreatic [50,51], thyroid [50], lung [52] skin [50,53], neuroblastic [54], and even blood cancer [55]. Here SCs generally act in a paracrine way to modify the immune microenvironment, promote tumorigenesis, and participate in the process of nerve invasion by tumour cells (perineural invasion (PNI) [56,57]. 

These exciting findings regarding extrinsic innervation prompted us to provide a comprehensive overview of known SC functions in the digestive system. 

## 3. Schwann Cell Involvement in Physiology and Disorders of the Digestive System 

In this section, we describe the functions of SCs and SC progenitors in the main organs of the digestive system. We summarise them in Table 1. Additionally, we briefly discuss the innervation of each structure, particularly if there is evidence for augmented innervation, which usually happens in cancer and inflammation settings. SCs themselves give rise to some tumours in the digestive system, such as schwannomas and SC hamartomas. These are out of the scope of this review since we focus only on the response of normal non-mutated SCs in various physiological and pathological situations.

### 3.1. Oral Cavity

The oral cavity is densely innervated by branches of the trigeminal nerve that has sensory and motor functions [84]. Chronic and acute orofacial pain conditions such as headache, and dental and cancer-related pain affect many people worldwide [85]. Oral cancers are among the 10 most prevalent cancers, have a poor prognosis, and are associated with intense pain [86,87]. The most common type of mouth tumours is oral squamous cell carcinoma (OSCC), which represents around 90% of all cases [88]. Loss of *TP53*, a gene most commonly mutated in head and neck cancers, further promotes innervation in mouse oral epithelia, while in patients with OSCC, increased innervation correlates with worse overall survival [89]. In fact, PNI is a common pathophysiological feature observed in up to 60% of OSCCs and increases the risk of lymph node metastasis [88,90].

Glia are recognised as mediators of orofacial pain [91,92], and several studies suggest that SCs are involved in the increased nociception [58,59] and nerve invasion [60,61] observed in oral malignancies. One of the proposed mechanisms is the activation of SCs in response to tumour necrosis factor alpha (TNFα) secreted by OSCC cells. This activation, in turn, stimulates TNFα and nerve growth factor (NGF) production by SCs themselves. TNFα overexpression is found in oral cancer tissues and is indeed correlated with elevated pain, while supernatants from activated SCs can increase facial allodynia in mice [59]. The TrkB/BDNF signalling axis is involved in the oral cancer PNI process, since modulation of this pathway in co-culture studies regulates SC—tumour cell interaction, and influences cell migration and differentiation [60,61]. Additionally, tooth SCs have a special status in dental pulp where they perform stem cell [22] functions, and modulate immune response [57] and nociception [92]. For a broader discussion of this topic, we refer the reader to a recent review [93].

### 3.2. Esophagus

The esophagus is innervated by many inputs, including vagal motor neurons, sensory neurons, and local enteric neurons [94,95]. Imbalance of inhibitory and excitatory neural activity [96] as well as sensitisation of esophageal afferent neurons by inflammatory mediators and endogenous substances (hydrogen, potassium ions, 5-HT, bradykinins, prostaglandins, etc.) [97], leads to various esophageal-related disorders.

Esophageal cancer is the sixth leading cause of cancer-related deaths worldwide [98]. The predominant types of esophageal cancers are squamous cell carcinoma (ESCC) and esophageal adenocarcinoma. These carcinomas have a strong tendency to metastasise, even if the tumour is superficial [99]. In a recent histological investigation of 260 human esophageal cancers, innervation in the tumour microenvironment was identified in 38% of the cancers and, more commonly, in ESCC, while signs of PNI were detected in 12% of the samples [100]. PNI was correlated with reduced survival [100], which supports previous findings that nerve invasion by tumour cells can be used as a prognostic factor in patients with esophageal cancer [101,102]. PNI was also found to be a better prognostic indicator than lymphovascular invasion in ESCC [103]. 

Several neurotrophic factor receptors, including Tropomyosin receptor kinase A (TrkA) [104], TrkB [105] and p75 [104,106,107] are expressed by esophageal tumour cells. A subset of esophageal tumour cells could express both NGF and NGF receptors [104], suggesting an autocrine signalling loop. Antigenic characterization combined with the genetic tracing of peripheral glia in murine esophageal tissue revealed the presence of myelinating and non-myelinating SCs of motor processes, a network of non-myelinating perisynaptic SCs, several types of enteric glial cells, and some glial cells along the blood vessels [95]. To what extent this rich variety of cells is involved in esophageal pathology remains to be demonstrated, but p75 expression is a hallmark of activated SCs, and glia-cancer cell interaction via p75 signalling has been reported in other gastrointestinal tumours [51]. Intriguingly, a portion of ENS neurons in the esophagus are derived from SCPs that travel along the vagus nerve [108]. This contribution occurs earlier than similar SCP-borne neurogenesis in the gut [27]; nevertheless, potential subsequent contribution of SCs to esophageal homeostasis and disease remains unexplored.

### 3.3. Stomach

The stomach is innervated by extrinsic parasympathetic vagal and sympathetic spinal nerves, as well as intrinsic neurons of the ENS [109,110]. A meta-analysis of the association between PNI and survival in patients with resectable gastric cancer concluded that PNI is an independent prognostic factor and can also serve as a predictive factor for tumour recurrence [111]. Additionally, PNI might help predict patients who could benefit from postoperative adjuvant therapy [112].

The role of innervation in gastric tumour has been studied in murine models. In an earlier report, myenteric denervation in rats using benzalkonium chloride resulted in a reduced incidence of chemically induced gastric cancer [113]. Similarly, denervation by vagotomy or botulinum toxin treatment in various mouse models of gastric cancer slowed disease progression and reduced tumour lesion incidence [114]. In these tumour models, neurons act by activating Wnt signalling through the muscarinic acetylcholine receptor M3 and thus promote the expansion of gastric epithelial cells. Stomach denervation leads to diminished Wnt signalling and reduced number of Lgr5+ epithelial stem cells. Furthermore, neurons in a gastric organoid co-culture system can substitute for the presence of mandatory Wnt3a in the culture medium [114]. 

The activation of the Wnt pathway in stomach epithelial cells was further studied later and attributed to tuft cell- and axon-derived acetylcholine. In a positive feedback loop, Ach-activated tumour cells produce NGF and recruit more Ach-secreting axons, leading to hyperinnervation and further Wnt activation [115]. Interestingly, in a genetic mouse line, where an excess amount of NGF is secreted by gastric epithelial cells, the stromal compartment of the lamina propria is significantly expanded, epithelial tissue architecture changes, and tumours arise. Among the stromal cells, the authors observed many Nestin+/s100b+ glial cells [115]. Given the ability of glial cells to secrete neurotrophic factors, including NGF, and participate in axon guidance, it would be intriguing to study in more detail the role of these cells in gastric tumorigenesis.

### 3.4. Pancreas

#### 3.4.1. Pancreatic Innervation and Insights from 3D Imaging 

Innervation of the pancreas has been studied for more than a century, with important findings and descriptions dating back to the age of major discoveries in anatomy and physiology [116]. Moreover, the most evidence for the involvement of SCs in the aspects of development, physiology, and disease of the gastrointestinal system comes from the studies of this organ. The pancreas is innervated by extrinsic and intrinsic neurons. Extrinsic are mainly associated with the vagus nerve and sympathetic splanchnic nerves, as well as sensory innervation, while intrinsic stem from intrapancreatic ganglia [4,117,118]. In addition, there are connections to the ENS [4]. Nerve endings can synapse at the ganglia or directly contact other pancreatic structures, for example, blood vessels, ducts, acini, and islets [118]. 

Recently, several studies have addressed pancreatic innervation in more detail using advanced tissue-clearing and imaging techniques [118,119,120,121]. Quantitative imaging of the adult mouse pancreas revealed that the nerve distribution in the organ is not uniform, with a larger volume of nerve seen closer to the duodenum. Depending on the neuronal marker used, up to 35% of all islets are contacted by the axons. These islets tend to be much larger so that around half of the islets by mass are innervated. Exocrine innervation is much less dense than in the endocrine portion and, similarly, is more prevalent in the duodenal region [120]. This agrees with earlier observations from electron microscopy studies, which noted dense innervation around arterioles and islets, and rather sparse innervation around the ducts and the acini of the murine pancreas [122,123]. 

The anatomically compact human pancreas is very different from the diffuse mesenteric type of pancreas found in murine species [124]. Differences are also seen with respect to innervation. Unlike in the murine pancreas, the innervation density in human tissue samples is similar in the endocrine and exocrine portions, while the proportion of innervated islets is smaller than in mouse tissue [120] and nerve-endocrine contacts are sparse [125]. Interestingly, the distribution of intrapancreatic ganglia was similar in humans and mice and was not significantly affected by diabetes [120]. 

As expected, species differences are also observed in the way SCs are distributed within the endocrine pancreas. Most of the Insulin+ islet mass in mice is contacted by axons [120]. Additionally, direct SC− endocrine cell contacts might increase the coverage even further. SCs form a dense envelope-like coating of murine pancreatic islets with SC processes reaching inside [123,126]. In human tissue, GFAP+ cells are less dense and are associated with autonomic neurons; however, some SC projections terminate at the endocrine cells [121]. SC processes associated with thin axons were also observed in the exocrine compartment together, forming a loose network on the surface of the acinar structures [122].

#### 3.4.2. Physiological Role of Innervation and SCs in Healthy Pancreas 

Autonomic regulation of pancreatic function, in general, is well described [127,128]. Parasympathetic activation induces the secretion of insulin and digestive enzymes, while sympathetic signalling results in reduced secretion and blood vessel constriction. Nevertheless, a detailed examination is revealing, and new functions of innervation come to light. For example, some sympathetic nerves project to the pancreatic lymph nodes and, if stimulated, can exert immunomodulatory functions [129]. Another recent study found that sensory innervation of the vagal branch makes extensive contacts with islets and β-cells use serotonin to communicate with these neuronal processes [130]. There may be more revealing studies to come since the choice of neuronal markers can be crucial for a correct assessment of the extent of innervation [120], especially given that axons can be as thin as 0.1 μM, while some of these markers are also expressed by parenchymal cells [122].

Insights from developmental studies suggest that innervation is critical for proper formation of the pancreas. Parasympathetic denervation leads to reduced β-cell proliferation in rats [131]. Disruption of sympathetic innervation and β-adrenergic signalling, on the other hand, results in hampered mouse endocrine cell maturation and altered insulin production [132]. This is reflected in the adult, since innervated islets in mice tend to be up to 10 times larger than the rest [120]. In addition, NC cells and glia appear to play a major role in this process. Migrating NC cells arrive in the developing murine pancreas soon after delamination and engage in reciprocal signalling with progenitors of the pancreatic epithelium to control islet mass [62]. As a result, β-cell proliferation is reduced and islet maturation is fostered (Figure 2A) [62,63]. Similarly, zebrafish NCs come into close contact with the endocrine epithelium before forming neurons that innervate the islets and pave the way for further neuronal contacts [64].

#### 3.4.3. Fate and Function of SCs in Disorders of the Endocrine Pancreas

Pancreatic innervation is clearly affected by the onset of endocrine disorders (Figure 2C–E). It was reported that in models of autoimmune type 1 diabetes (T1D), such as non-obese diabetic (NOD) mice, general and sympathetic innervation is reduced [65,133]. On the contrary, other studies suggested increased innervation of surviving islets in tissue samples from both streptozotocin (STZ)-treated and NOD mouse pancreas [120,134]. Similar observations were made in human samples, and even more so, axon–endocrine cell contacts were preserved in samples from patients with type 2 diabetes (T2D) [120]. The same is true for the glial counterpart of innervation. After experimental STZ treatment, which models T2D, and in early insulitis in NOD mice, reactive gliosis is observed around the islets [65,66,67]. After STZ injury, SCs demonstrate significant outgrowth and make more membrane contacts with the intra-islet capillaries [68]. With progression of insulitis, SCs die prior to β-cells [135]. One of the hypotheses suggests that peri-islet SCs could act as antigen presenters at the onset of T1D and amplify inflammatory signals [67]. 

The observation of glial cells surrounding the islets of Langerhans [123] and NC cells in close contact with developing endocrine cells [62] could have contributed to the idea of improving islet grafts by co-transplantation with NC stem cell (NCSC)-like cells (Figure 2B). Unlike in developmental settings in vivo, where NC cells limit endocrine cell proliferation [62,63], co-transplantation with NCSC spheres promotes adult β-cell expansion [70]. Insulin production by co-transplanted islets is also increased and can partially restore normoglycemia in mice after heterotopic transplantation [70]. Similarly, transplantation from human islets with NCSC spheres SCs mouse embryonic DRGs promotes endocrine cell proliferation, vascularisation, and innervation of the graft [71]. The same group developed a method to coat the surface of murine islets with NCSCs before transplantation. This improved engraftment into liver tissue, islet vascularisation, and overall performance of the graft. Moreover, many NCSCs migrated into the graft and differentiated to glial and neuronal phenotypes [72]. Interestingly, donor SCs are normally present in the grafted islets, and the fate of these cells was studied in optically cleared mouse islets after transplantation under the kidney capsule. SCs survive transplantation and appear to be the main contributors to the re-established SC network; however, this process is slow and does not reach the extent seen in in situ islets [136]. Along these lines, NCSCs in co-culture were able to protect insulin-producing islet cells from cytokine-induced death, suggesting a potential immune-modulatory action [73]. Taken together, SCs remain in the grafted islets, but the addition of glial cells or NCSCs could be beneficial. However, caution should be taken with such approaches, as SCs might actively participate in the immune process and exacerbate autoimmune response [67,69].

#### 3.4.4. Innervation and SCs in Disorders of the Exocrine Pancreas

The exocrine pancreas makes up the largest portion of the organ, or about 95% by mass, and is the site of origin of various pancreatic disorders [137]. Pancreatitis, cystic fibrosis, and cancer are just a few disorders that affect the function of this tissue. Pancreatic cancer is known for its grim prognosis at the time of diagnosis and has the lowest survival rate among cancers in Europe [138].

Both pancreatitis and cancer are linked to abnormal innervation. In fact, nerve size and innervation density correlate well with the severity of pancreatic disease and increase as the tissue progresses from normal to inflamed and malignant [139,140,141]. Neuroinflammation, induction of neuronal plasticity, changes in the type of innervation and supporting cells have all been documented in disorders of the exocrine pancreas and could contribute to pancreatic neuropathy [141,142,143]. One of the common alterations that accompany such neuropathic changes is PNI, which develops in virtually all cases of pancreatic ductal adenocarcinoma (PDAC) and correlates with increased morbidity and pain [144,145].

An effort has been put into deciphering the roles of various nerve branches in pancreatic cancer. For example, sensory innervation increases in cancer, explaining the pain sensation associated with this condition [143]. Ablation of sensory innervation can delay the onset and progression of PDAC, possibly through the action of substance P [146]. Compelling evidence suggests that disruption of parasympathetic and sympathetic neuronal homeostasis can also have dramatic effects. Prolonged stress and an increased level of catecholamines can accelerate the progression from neoplasia to PDAC. In this case, stimulation of β2-adrenergic receptors on tumour cells makes them secrete NGF which, in a feed-forward loop, attracts even more axons and demonstrates the pro-tumorigenic function of the sympathetic nervous system [147]. The parasympathetic branch of the ANS apparently acts oppositely. Thus, vagotomy in a mouse pancreatic cancer model promotes tumour growth and reduces survival through TNFα [148]. In line with these findings, cholinergic innervation has been shown to suppress pancreatic tumorigenesis [149]. On the other hand, there is a conflict of data regarding the role of autonomic innervation. A decreased amount of sympathetic fibers has been reported in pancreatic tumour tissue [141] and one study actually revealed a protective immunomodulatory action of sympathetic nerves in non-stressed conditions [150]. Cholinergic signals can also have immunosuppressive effects and promote PDAC progression. In the case of strong PNI, increased levels of acetylcholine change the immune microenvironment of the tumour to a less immunoprotective one by decreasing the activity of CD8+ T cells and favouring the type-2 T helper cell phenotype [151].

In addition to neuron-mediated effects, other contributions of stromal cells to pancreatic neuropathy have been extensively studied. Immune cells, cancer-associated fibroblasts, and SCs have all been implicated (Figure 3) [152]. PNI in pancreatic cancer is not fully understood, but is likely driven by an initial interplay of signals from tumour and immune cells, gradually accompanied by injury response cues from various activated nerve-associated cells [56]. An early report suggested that adenocarcinoma cells interact with SCs in PNI, as apparent from patient tissue sections [153]. Another immunohistological study of human tissue reported an activated state of intrapancreatic glia in pancreatitis and pancreatic cancer patient samples, as evidenced by increased Nestin expression [141]. Later, it was shown that SCs are localised in the vicinity of neoplastic pancreatic and colon lesions before PNI. SCs displayed tropism to pancreatic tumour cells, at least in part dependant on the NGF-p75 signalling [51]. This was a novel development and suggested that in PNI settings, nerve components could be the first to migrate. The same research team proposed that SCs are activated through various routes, notably through hypoxia, tumour-derived interleukin (IL)-6, and chemokine CXCL-12, overexpressed in the pancreatic intraepithelial neoplasia (PanIN) [74,75]. In this scenario, activated SCs participate in pain suppression in early PDAC lesions by modulating the activity of the astroglia and microglia of the spinal cord and could cause a delay in the diagnosis of the disease. Demir and colleagues showed that glia-specific inactivation of the CXCL-12 receptor CXCR4/CXCR7 or blockade of IL-6 signalling abrogated SC migration, decreased glial cell numbers in PanIN lesions, and increased pain sensation in mice [74,75]. The hypoxic environment was also found to activate SC migration because glial cells responded to granulocyte-macrophage colony-stimulating factor (GM-CSF) produced by PDAC cells. GM-CSF expression was elevated in patient tumour samples and was correlated with PNI incidence [76].

To complicate things, both CXCL-12 and IL-6 can also be secreted by SCs [74,77,154]. In a loop of cytokine signalling, IL-6 from SCs through the STAT3 pathway triggered tumour cell EMT and migration, while tumour-derived IL-1β activated the NF-kappa B pathway in glia and further induced SC cytokine secretion from SCs [77]. The same study directly correlated the increased presence of SCs in the patient tumour stroma with worse survival. On the contrary, another group looked at patient sections and used a co-culture approach to link SCs at PNI sites with PDAC mesenchymal to epithelial transition, a process tumour cells use to establish metastasis [155]. In addition to the IL-6/IL6R pathway, upregulation of STAT3 signalling in PDAC cells through another route has been implicated in nerve invasion. In this case, the SC-secreted soluble L1 cell adhesion molecule (L1CAM) serves as a chemoattractant for tumour cells and leads to increased production of matrix metalloproteinase MMP-2 and MMP-9, which help in PNI process [78]. Interestingly, high expression of L1CAM in pancreatic tumour tissue samples has previously been associated with increased PNI and pain [156]. Furthermore, soluble factors found in the conditioned medium of the commercial SC line S16 increased the cleavage of another adhesion molecule integrin A6B1 to the more motile A6pB1 form. The authors demonstrated that tumour cells with cleaved integrin form can invade laminin more easily and suggested that the enzyme urokinase-type plasminogen activator from SC-conditioned medium could perform the cleavage [157].

Autophagy, a process that activated SCs use in nerve repair to reduce cell membrane size and clear myelin debris [158], is also implicated in PDAC [82]. Murine PNI and cell culture models were used to demonstrate that tumour cells produce NGF, which acts through ATG7 (activated in autophagy 7) to induce autophagy in SCs and promote invasion of the nerve.

SCs can be activated or participate in PDAC tumorigenesis indirectly by communicating with other cells of the stroma. For example, by producing CCL2, SCs attract CCR2-expressing monocytes that eventually differentiate to tumour macrophages and promote nerve invasion [79]. An axon guidance molecule SLIT2 derived from tumour fibroblasts modulates N-cadherin/b-catenin pathway to induce SC proliferation and migration and promote neurite outgrowth [80] Likewise, tumour stroma-derived leukemia inhibitory factor (LIF) activates STAT3 signalling in SCs, leading to SC differentiation and increased neuronal remodelling in PDAC [81]. 

Together, these studies demonstrate a complex context-dependant paracrine function of SCs in the stroma of pancreatic tumours and warrant further exploration, for example, to establish whether the dispersed SCs found in the tumour stroma are associated with the axons or are independent and which exact nerves contribute SCs to the tumour microenvironment. A more thorough investigation, involving 3D imaging of normal and tumour sites in mouse and human tissue, similar to that performed in the endocrine pancreas, might answer these questions [118,119]. Importantly, SC involvement is apparent already in early pre-cancerous lesions and present an opportunity for potential intervention [51,74,75]. Until now, 3D imaging of PanIN lesions allowed visualisation of glial activation in an early stage of tumorigenesis and demonstrated a 5-fold increase in GFAP+ fibers [159]. Furthermore, many soluble factors of neurotrophic or axon-guiding nature such as NGF [147,160], brain-derived neurotrophic factor (BDNF) [147,161], neurotrophin 3 (NT-3) [147,162], glial-cell-derived neurotrophic factor (GDNF) [163], neurturin [164], artemin [165,166], sonic hedgehog [167,168], SEMA3D [169], and others are overexpressed by pancreatic tumour cells. SCs are well known to express the same factors in injury settings [43]. Additionally, other cells that form the pancreatic tumour microenvironment, for example, macrophages [170] and cancer-associated fibroblasts [80,171], can provide factors that are normally expected to come from glia or neurons. Thus, careful cell-specific ablation and functional tests are needed to delineate the source and direction of such signals.

### 3.5. Gall Bladder and Bile Ducts

The gall bladder receives sympathetic, parasympathetic, and sensory innervation with major input provided by the vagus nerve [172]. According to a surgical anatomy study of human samples, innervation occurs predominantly through the anterior and posterior hepatic plexuses. Additionally, input from phrenic nerves and extrinsic neural connections with the duodenum are observed [173]. Unlike the liver, which has no intrinsic innervation [174], the gall bladder contains its ganglia located within the walls of the organ that integrate signals from the brain, the sympathetic and enteric ganglia, and the sensory fibers. These well-developed autonomic ganglia share embryonic origin with the neurons and glia of the ENS; however, a study of these plexuses in the gall bladder of guinea pigs suggested some major structural and physiological differences [175]. These networks of extrinsic and local innervation regulate organ physiology, for example, smooth muscle and epithelial cell function, and are implicated in disorders of the biliary tract. 

Cholangiocarcinoma (CCA) is a rare aggressive cancer with limited response to chemotherapy that affects the bile ducts outside or within the liver [176]. These tumours are sometimes referred to as neurotropic [177] and show a high frequency of PNI—in up to 75% of extrahepatic [178] and 29% of intrahepatic forms of CCA [179]. Tumour cell invasion into nerves in CCA is correlated with a decrease in overall 5-year survival rates [180] and can serve as an independent prognostic factor for intrahepatic forms of CCA post-resection [181]. Shen and colleagues summarized the CCA nerve invasion-related molecules and suggested that acetylcholine, NGF, neural cell adhesion molecule (NCAM), MMPs, and tissue growth factor (TGF) could have prognostic relevance [182]. 

Gall bladder cancer is another rare biliary tract malignancy with frequent PNI that has a prognostic impact and correlates with the late stage of the disease. In this case, tumour cells use the neural route to invade tissues surrounding the extrahepatic bile ducts [183]. The role of SCs in the gallbladder and its pathologies has not been extensively studied.

### 3.6. Liver

The parasympathetic and sympathetic nervous system provides extrinsic innervation of the liver through the portal area. Sympathetic innervation comes from the celiac and mesenteric ganglia, while parasympathetic innervation is represented by branches of the vagus nerve stemming from the dorsal motor nucleus, which can either directly reach liver tissue or synapse at the hepatic plexus [184]. 

These afferent and efferent nerves gain access to liver tissue along the portal vein, hepatic artery, and bile ducts. In humans, unlike rodents, sympathetic innervation reaches deeper into the liver lobules along sinusoids and connects with the parenchymal cells [184]. Hepatic innervation differs from that of the gastrointestinal tract, although the liver is formed from the foregut during embryonic development. The liver itself does not contain ENS or other intrinsic neurons derived from NC, perhaps due to very low expression levels of GDNF, a known NC chemoattractant in the developing gut [174,185]. 

Autonomic innervation plays an important role in liver physiology and homeostasis. Parasympathetic impulses regulate nutrient sensing, bile production, regulation of homeostasis, and response to toxins. Sympathetic innervation ensures the release of noradrenaline, which is involved in glycogenolysis. Innervation also plays a critical role in liver regeneration processes, food intake, hepatic blood flow, and circadian rhythm regulation [186,187]. Aberrations of normal liver innervation can lead to liver disorders. Stimulation of the sympathetic nervous system has been shown to promote hepatic fibrosis and decrease liver regeneration [188]. Cholinergic denervation attenuates liver chemical damage by down-regulating liver stellate cell activity and decreasing the levels of fibrogenic cytokine TGF-beta1 and bone morphogenic protein (BMP) 6 [189]. During non-alcoholic fatty liver disease, innervation is disorganised and degenerates proportionally to the severity of the damage [190]. 

Homeostasis of the liver is regulated by the interaction of cytokines, growth factors, and signal transducers secreted by Kupffer cells and other cell types. Coordination of these factors within various cells and even organs is involved in the hepatic regeneration process. Various roles of the autonomic nervous system have been described in this process. [191]. For example, vagal nerve-released acetylcholine induces the production of IL-6 by liver macrophages, which activates hepatic FoxM1 expression and hepatocyte proliferation [192]. At the inter-organ level, serotonin has been shown to be released by platelets and enterochromaffin cells and contribute to liver regeneration by induction of hepatic cell proliferation [193]. 

Despite the accumulating evidence on the importance of liver innervation in regeneration and disease, the role of glial cells in these processes has not been studied so far. Scarce data are limited to observations of SCs in association with unmyelinated nerve fibers in the Disse space close to perisinusoidal cells in human liver tissue specimens [194] and in transplanted liver tissue in rats [195].

### 3.7. Bowel

The intestines are constantly exposed to large amounts of foreign material that has to be sensed and monitored [196]. The ENS as part of such a control system is very elaborate in the gut and can even function separately from the CNS. However, in general, ENS is very integrated and has many connections to the CNS, the major branches of the PNS, and local neuronal plexuses in digestive organs such as the pancreas and gall bladder [5]. Based on location, morphology, and molecular profiles, up to six subtypes of enteric glia are identified in the gut [7]. Sox10, Plp1, S100b and GFAP are useful markers that help to identify enteric glia, but these proteins are not expressed by all glial subtypes and hint at the plasticity of the different phenotypes. Indeed, this was well-highlighted in a recent study that showed that GFAP+ cells arise from Plp1+ glia and support the intestinal stem cell niche by providing Wnt ligands [197]. The authors demonstrated that ablation of the GFAP+ population in mice leads to a transient disruption of epithelial homeostasis and that the GFAP+ cell pool is replenished by Plp1+ cells. Furthermore, GFAP+ glia support intestinal regeneration in a paracrine way [197] and thus could represent an ‘activated’ state of the ENS glia, similar to the activated SCs seen in regeneration of the skin and digit tip [44,45,47].

In addition to ENS neurons, the glia are involved in virtually all basic aspects of gut homeostasis and disease, including motility disorders [198,199,200], inflammation [201,202], infection [203,204,205], and tumorigenesis [206,207], as well as extra-intestinal neurologic disorders [208,209,210]. The profound involvement of enteric glia in the coordination of these processes could be explained by the formation of neuro-glial units [211,212] and constant crosstalk with the immune components of the intestinal cellular environment [213]. Two recent extensive reviews covered the multitude of functions of ENS glia in detail [7,38]. In this section, we summarise the information regarding SC functions in the small and large intestine. 

In addition to intrinsic innervation, ensured by ENS, mammalian gut function also depends on extrinsic innervation. The small intestine consists of the duodenum, jejunum, and ileum, and functions to breakdown and absorb nutrients. It is extrinsically innervated by the parasympathetic and sympathetic nervous system. Parasympathetic fibres of the vagus nerve are responsible for the control of secretions and motility, while sympathetic fibres of splanchnic nerve ganglions control blood vessels, secretions, and motility [214]. The large intestine is composed of the cecum, appendix, colon, rectum, and anal canal, and is mainly involved in the absorption of water from indigestible contents. Midgut structures such as the ascending colon and proximal two-thirds of the transverse colon are extrinsically innervated by parasympathetic, sympathetic and sensory nerves from the superior mesenteric plexus, while the distal third of the transverse colon, the descending and sigmoid colon are innervated from the inferior mesenteric plexus [215]. SCs in the gut, unlike enteric glia, are associated with extrinsic innervation. Nevertheless, the markers mentioned above used for enteric glia can also be expressed by SCs and could make it difficult to distinguish which cells exactly contribute to the observed effects [7].

ENS of the gut develops from the progenitors of vagal and sacral NC that colonise the developing bowel [216]. However, a portion of the neurons and glia in the gut arise from Sox10+ SCPs that can be traced in *Dhh-Cre* mice [2,27]. SCP neurogenesis is minor in the small intestine, but in the mouse colon up to 20% of ENS neurons can be derived from SCPs. This contribution is functionally substantial since the loss of *Dhh-Cre-traced* neurons causes oligoganglionosis. Similarly, ongoing post-embryonic neurogenesis by extrinsic SCPs takes place in zebrafish [83]. In some pathological conditions such as colitis, adult ENS is abnormal and hyperinnervation can be observed, raising the question of adult neurogenesis [217]. Indeed, adult neurogenesis in mice and zebrafish can occur in response to chemical stimulation and injury [83,218,219]. Therefore, it is possible that Sox10+ SCs of extrinsic nerves could participate in this neurogenesis, especially since BrdU-incorporating cells migrate from niches outside of the ENS ganglia [218]. Recently, another study found that a population of ENS glia that does not express canonical glial markers performs neurogenesis in the gut of adult zebrafish under homeostatic conditions [220]. Together, this suggests that there may be several sources of new enteric neurons in mammals and teleosts.

In Hirschprung disease (HSCR), due to impaired progenitor migration, colon segments remain underpopulated by ENS progenitors and lack ENS ganglia, eventually causing obstruction and inflammation of these areas. However, hypertrophic extrinsic innervation of aganglionic segments containing many SCs is a hallmark of HSCR [221]. Soret and colleagues demonstrated that in several mouse models of HSCR, GDNF treatment in the neonatal period can induce gliogenesis and neurogenesis, improving gut motility and prolonging survival. This recovery, at least in part, is due to SCs that function as stem cells in such settings [28]. In another HSCR mouse model study, traced SCPs could partially compensate for reduced neurogenesis in the aganglionic colon and small intestine, suggesting that SCPs can perform robust neurogenesis irrespective of the bowel segment [26]. Importantly, this neurogenesis was limited to the transition zone and did not occur in the aganglionic region itself, suggesting that the presence of at least some intrinsic neurons is important for it to take place [26]. Furthermore, human SCPs derived from hypertrophic nerve bundles of HSCR-affected intestinal segments, upon transplantation into the aganglionic mouse colon, could differentiate into glial and neuronal phenotypes [222]. Collectively, these studies suggest that SCs associated with extrinsic gut innervation are a potential target for therapies that aim to restore ENS in the gut.

Patients with inflammatory bowel diseases (IBD), such as ulcerative colitis and Crohn’s disease, are exposed to chronic inflammation of the colon or other segments of the gastrointestinal tract. Enteric glia are actively involved in neuroimmune modulation, and many studies, sometimes with contrasting results, have linked ENS glia with IBD. In ulcerative colitis, inflammation-activated glial cells trigger neurotoxicity and cause neuronal death [223]. On the other hand, colitis promotes neurogenesis through the apparent differentiation of Sox2/Plp1+ glia into excitatory neurons [205]. More studies find anti-inflammatory action through the production of lipid metabolites that could improve the intestinal epithelial barrier in Crohn’s disease [224,225]. To our knowledge, the function of glia associated with extrinsic innervation has not been adequately studied in IBD. However, it is possible that tissue damage caused by inflammation can recruit SCs, which then would participate in immune modulation [226] or perform neurogenesis [227].

## 4. Concluding Remarks

### 4.1. Open Questions and Directions

Even with the recent development of the field, many questions remain unanswered. For example, the extent and type of innervation in organs of the digestive system is not yet completely mapped, with some contrasting reports [133,134]. Advanced optical clearing and imaging techniques, similar to the ones used in the pancreas [119,120] and the liver [190], are extremely useful. However, depending on the assay or marker used, large discrepancies can be observed [120], perhaps due to sample preparation, the minute size of the axons, and the low quantities of the target protein [118]. In addition, the source of glial cells in these organs and various conditions remains elusive. More precise, single-cell sequencing data-informed, immunohistological detection combined with tracing and 3D-imaging approaches could help to advance this direction.

Despite a clear morphological distinction between ENS glia and myelinating SCs [38], they share many transcriptional similarities [228]. Thus, it is not clear to what extent SCs associated with the extrinsic innervation of the digestive system would be different from ENS glia in the activated state—after injury or in cancer settings. The fact that a fraction of ENS cells originate from SCPs and can be substantial (up to 55% of cecum neurons are of SCP origin in mice [26]) can further blur the boundaries and complicate the distinction. Similarly, there is not much known about the heterogeneity of nerve-associated SCs themselves and how distinct the activation mechanisms for these cells would be. The SCPs that the SCs originate from can be of a distinct embryonic origin [17], and it is conceivable that the SCs in various anatomical locations are different. Influenced by the local tissue environment, these cells would respond to injury or disease onset differently. Indeed, recently a single-nuclei sequencing approach was used to profile sciatic, sural, peroneal, and vagus nerves, and several new subtypes of myelinating and non-myelinating SCs were identified [31]. A similar survey of other types of NC-derived glia, such as the terminal SCs in the neuromuscular junctions or non-myelinating SCs associated with fibers of smaller calibre would be enlightening. Furthermore, single-cell sequencing studies of nerve-associated cells in various organs in normal vs. injured or tumour settings will help to address more unknowns [229,230]. 

Based on the publications we reviewed, several translational directions appear. One is clearly related to the functions of SCs in carcinomas, such as the promotion of PNI, immune modulation, and mediation of pain signals. Preventing SC activation or targeting SCs to limit paracrine stimulation of the tumour microenvironment could be one possible strategy, especially for tumours with a high prevalence of PNI, such as PDAC. This could be achieved by interrupting SC communication with neuronal, tumour or immune cells, for example, using neurotrophic factor or TRK inhibitors that could target several of such interactions [231,232,233]. Many parallels exist between SC activation mechanisms in nerve injury and tumour settings, so targeting some of these mechanisms to promote the differentiation state of SCs could be feasible [234]. Pain is a major morbidity factor in orofacial cancers, and inhibition of TNFα [59] or TNFα downstream targets involved in SC activation, such as c-jun N-terminal kinase (JNK) or NF-kB, deserves further investigation. Additionally, targeting pathways involved in neuropathy could be considered [235,236].

In intestinal inflammatory disorders, the promotion of the neurogenic potential of SCs is an exciting strategy, as exemplified by the successful use of GDNF enemas in mouse models of HSCR [28]. There are several compounds that can target the SC de-differentiation pathway and promote activation or induction of the repair phenotype in SCs, such as neuregulin isoforms, sphingosine-1-phosphate receptor agonists, or even paclitaxel [234,237]. However, it remains to be demonstrated that such SC activation alone is sufficient to trigger plasticity and neurogenesis in the gut. Other approaches to stimulate endogenous regeneration potential of SCs could be tested, including controlled nerve injury that could activate SCs in the distal portion of the nerve. Interestingly, most of the neurogenesis induced by GDNF in mouse models of HSCR is not due to *Dhh*-traced SCs, suggesting the existence of another neuron progenitor [28]. A better understanding of glial cell heterogeneity and plasticity could allow addressing limiting factors for future clinical applications, for example, in the context of HSCR, where SCPs preferentially generate new neurons in a transition zone rather than in the aganglionic area itself [26]. Progenitor cell transplantation in such areas is a direction currently being investigated, and SCs should be considered as potential candidates [238].

Another direction is to improve the islet grafts for transplantation. Here, harnessing the original glial cells in the islets for autotransplantation or incorporation of glial components into pluripotent stem cell (PSC)-derived islets could be among the possible strategies. This could potentially promote revascularisation and reinnervation of grafts and improve islet performance [72].

### 4.2. In Vitro Approaches to Advance the Field

In the sources we reviewed, the use of co-culture approaches and conditioned media is commonplace. SCs have strict culture requirements and rely on constant trophic support and contact with the ECM substrate. Cells of various sources, including commercial immortalized, primary and induced, can be used, but primary SCs from a cognate PNS source would be advantageous [239]. Human SC cultures could be laborious to derive and propagate for extended periods, but there are reliable methods that allow them to be established for at least a limited use [240]. 

Cultured glial cells acquire a different transcriptional profile [229], and the same is true for other primary cells, so the use of more advanced 3D culture approaches could be beneficial. Co-culture in ECM gel is used to better model the interaction of neurons and SCs with epithelial cells [51] and larger structures, such as endocrine islets [72,132]. Another possibility could be the use of tissue-specific organoids. A recent consensus paper has suggested classifying organoids as epithelial, multi-tissue and multi-organ [241]. Epithelial organoids do not contain cells of mesodermal or ectodermal lineages as an integral part but can be temporarily co-cultured with such cells. This would require adjustments in the culture conditions, but it is feasible and has been performed, for example, with mesenchymal cells [242] and neurons [147,149,243,244,245]. Multi-tissue organoids, on the other hand, could be developed from aggregated precursors of distinct lineages and engineered to form a functional synchronized structure [246,247]. 

This has been achieved for gut ENS by differentiation of PSCs under defined conditions that produce gut organoids with glial cells [248]. In another approach, the PSC-derived ENS precursor and intestinal epithelial cells are combined to form multicellular organoids [249,250]. To further advance this concept, the precursors of all three germ layers have been combined to form stomach and esophageal organoids with an innervated smooth muscle layer [251]. Alternatively, engineered scaffolds are populated by primary or PSC-derived cells to create innervated mini-intestines [252,253,254]. Outside of the digestive system, sensorimotor organoids with functional neuronal components have been created from neuromuscular progenitors [255,256] or in a gastruloid approach to study the developmental process [257]. For extrinsic innervation, other approaches could be beneficial, for example, in combination with microfabricated devices, as described for the peripheral nerve [258]. Co-culture with organoids can serve multiple goals. It can be seen as a way to model the interaction between cells, but it can also be a tool to promote the maturity of the test system and to make it more relevant for use in regenerative medicine. However, the complication of the models may come with a risk of reproducibility [259].

### 4.3. Conclusions

Having researched the recent literature regarding the functions of extrinsic innervation and SCs in the organs of the digestive system, we find diverse SC roles described particularly in the mouth, the pancreas, and the intestine. We note that most of the scientific evidence concerns the involvement of SCs in cancer. 

Most of the research data comes from SC studies in the pancreas, where glial precursors and glia are involved in the development of the endocrine pancreas and are observed in association with the cells of healthy endocrine islets and acini and become ‘activated’ in disorders of both the endocrine and exocrine pancreas. In diabetes, the number of SCs expands and their functions have yet to be established, but they may interact with cells of the immune system. In a pancreatic tumour, SCs modulate the immune response and perception of pain, and promote tumour spread by attracting tumour cells to the nerves. 

Similarly, in the oral epithelium and salivary glands, SCs mediate pain and promote PNI. Taking into account the parallels between the mechanisms of carcinoma progression and propensity to invade nerves in other parts of the digestive system (e.g., in the esophagus and stomach), similar functions of SCs could be expected there.

In the gut, SCPs give rise to ENS neurons and glia. In some species, these functions persist into adulthood where, apparently, extrinsic nerve SCs can act as stem cells to replenish ENS. Several mechanisms of action of SCs were identified and could be used for therapeutic interference (summarised in Table 1). Additionally, there are at least two avenues for a possible clinical application of SCs. One in the form of a supportive cell for the transplantation of endocrine islets to improve the grafting process and one in gastrointestinal neuromuscular pathologies, such as HSCR.

## Figures and Tables

**Figure 1 cells-11-00832-f001:**
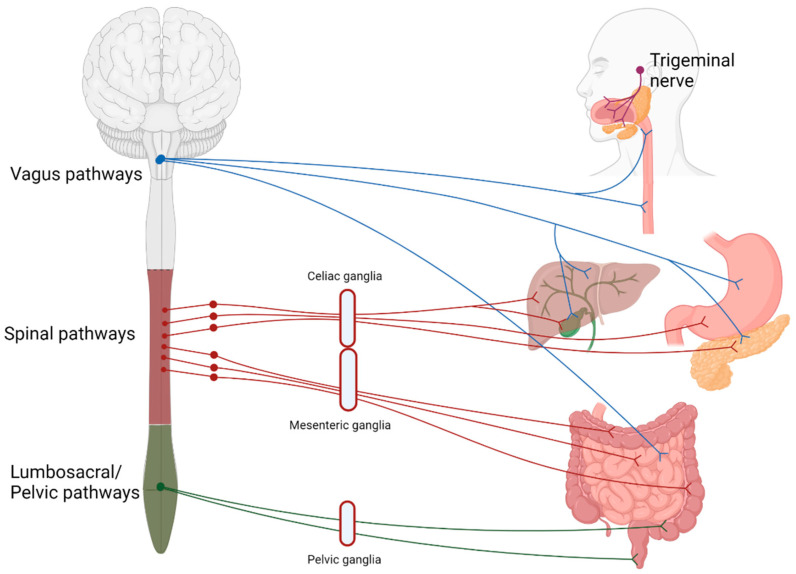
Schematic representation of the extrinsic innervation of the digestive system. Sympathetic innervation is performed mainly by spinal nerves (red), while parasympathetic by the vagal (blue) and pelvic branches (green). Both the spinal and vagal pathways are also involved in sensory and motor innervation. The trigeminal branches (purple) serve primarily sensory and motor functions.

**Figure 2 cells-11-00832-f002:**
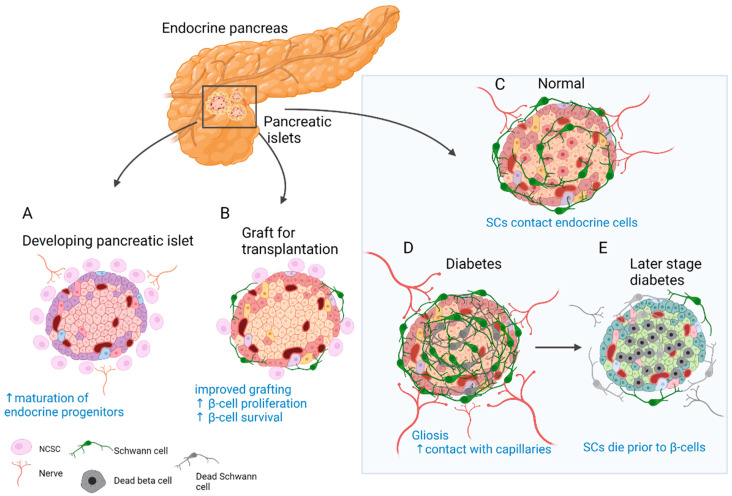
Role of SC and neural crest (NC) cells in the physiology and disease of the endocrine pancreas. (**A**) Reciprocal signalling with endocrine progenitors promotes islet maturation and glial fate choice by NC cells. (**B**) The transplanted islets contain surviving donor SCs. Co-transplantation and coating of endocrine islets with NC stem cell (NCSC)-like cells improve graft function and stimulate β-cell proliferation and survival. (**C**) SCs cover the surface of endocrine islets in mice (to a lesser extent in humans) and make contact with endocrine cells. (**D**) In T2D and insulitis, SCs expand and gliosis is observed. Extensive contacts are detected with capillaries. (**E**) With the onset of T1D SCs die before β-cells and innervation is reduced. SCs might have immunomodulatory function.

**Figure 3 cells-11-00832-f003:**
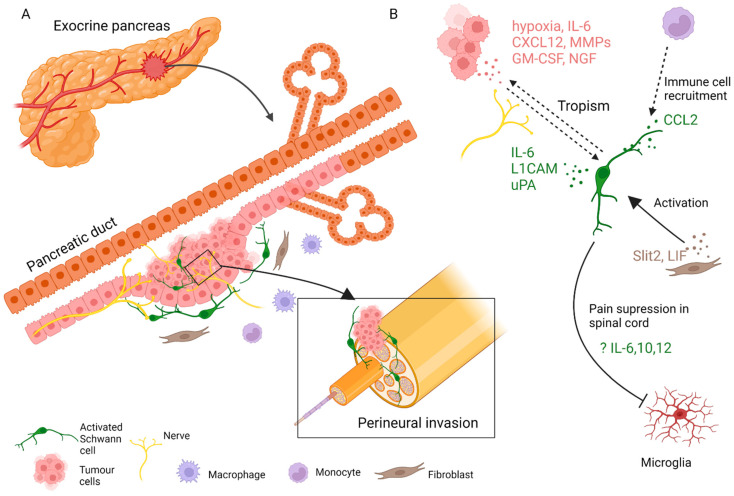
SC involvement in the pathophysiology of pancreatic cancer. (**A**) SCs, fibroblasts, immune cells, and other cells of the stroma contribute to the progression of pancreatic tumours and nerve invasion. (**B**) SCs engage in paracrine interaction with tumour and stromal cells.

**Table 1 cells-11-00832-t001:** Functions of Schwann cells (SCs) and SC progenitors in the organs of the digestive system.

Organ/Process	Observation/Mechanism	SC/Progenitor Specific Action
**Mouth**		
Oral cancer	Tumour activated SCs mediate pain [58,59]	TNFα, NGF [58,59]
	SCs promote EMT (possible link to PNI) [60]	BDNF [60,61]
**Pancreas**		
Development	Reciprocal signaling with endocrine progenitors promotes islet maturation and glial fate choice [62,63] NCs contact islets and perform neurogenesis [64]	Direct cell contact [63,64]
Diabetes	SCs expand around islets in insulitis and T2D [65,66,67,68]	Direct contact with capillaries [68], possible immune modulation [69]
Graft Optimisation	Co-transplantation with NCSCs increases β-cell proliferation and improves graft function, protects β-cell [70,71,72,73]	Direct cell contact [73]
Pancreatic Cancer	Tumour attracts SCs through IL6 or CXCL-12/CXCR4/7 pathway. SCs act on microglia to suppress pain and promote PNI [74,75]	IL-6,10,12, VEGF, G-CSF [74]
	Hypoxia-induced GM-CSF activates SCs and forces their migration to promote PNI [76]	Possible role in migration
	SCs activate the STAT3 pathway in PDAC and promote EMT [77] and MMP production [78] by tumour cells	IL-6 [77] and L1CAM [78]
	SCs influence tumour microenvironment by recruiting macrophages and promoting PNI [79]	CCL2
	Tumour fibroblasts activate SCs via N-cadherin/b-catenin and STAT3 pathways to promote neural remodelling [80,81]	Possible roles in differentiation and migration
	PDAC cells activate autophagy in SCs via NGF/ATG7 pathway [82]	Autophagy
**Bowel**		
Regeneration	SCPs perform neurogenesis in post-embryonic small [26,27] and large intestine [26,27,28] of mice and in zebrafish [83]	Differentiation to ENS neurons and glia
Colon cancer	SCs observed in the vicinity of neoplastic colon lesions [51]	Not described

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
