# Peer review of "Schwann Cells in Digestive System Disorders"

_cells, 2022, doi:10.3390/cells11050832_

Round 1

Reviewer 1 Report

1) General comments

Dr. Goluba and Dr. Parfejevs, et al. reviewed ‘Schwann cells in digestive system disorders’. This article is well presented. The reviewer has some comments.

  1. The authors have reviewed over 200 references. However, the reviewer thinks the focus of this article is unclear. How the authors would like to describe in this article about Schwann cells in digestive system disorders, such as existence, function of Schwann cells or the association between carcinoma and pain or invasion. The reviewer would like to know how this paper can contribute to clinical practice.
  2. The authors described ‘4.1. Open questions and directions’, ‘4.2. In-vitro approaches to advance the field’ and ‘4.3. Conclusions’ in ‘4. Concluding remarks’. The reviewer thinks these are really important for review. The reviewer recommends that the authors should focus on these Concluding remarks and review them.

Author Response

We thank the reviewer for the positive assessment of our work and valuable comments.

 Reviewer #1, Point 1:

“ The authors have reviewed over 200 references. However, the reviewer thinks the focus of this article is unclear. How the authors would like to describe in this article about Schwann cells in digestive system disorders, such as existence, function of Schwann cells or the association between carcinoma and pain or invasion..”

Reviewing Schwann cell involvement in digestive system disorders is not trivial because both digestive system is diverse and the functions of Schwann cells are not sufficiently studied. Schwann cells in digestive system are mostly associated with extrinsic innervation, which prompted us to include more general information on the innervation of the organs,

  Indeed, as the reviewer points out, activation and proliferation of Schwann cell is often seen in association with pain and nerve invasion by tumour cells in carcinomas that affect digestive system organs.

We have tried to make the focus of the paper more clear by changing the introduction. We included more information on SC development and changed the wording to be more direct. For example, we replaced  a more broad term “glia” in the introduction with more specific “Schwann cell”, where it was appropriate.

“..The reviewer would like to know how this paper can contribute to clinical practice.”

We are glad that this point was raised since finding actionable targets for therapies is what motivates much of the research. The original submission contained a table that summarized the molecules potentially important for Schwann cell action in digestive system. To expand on that, we used that information and included a discussion on what we consider the possible translational directions.

Reviewer #1, Point 2:

We appreciate this reviewer’s very valid suggestion to provide more discussion in the final sections of the manuscript. We have reviewed these sections and introduced some changes, including references to more recent findings in the field and a discussion on the translational potential of the generated knowledge.

The Reviewer has suggested style editing of the manuscript. We have performed it using Grammarly and Writefull Revise service.

Reviewer 2 Report

The present manuscript is dedicated to the tole of the PNS in the digestive organs and its relationship to digestive diseases, in particular cancer. This review is clear and very informative, showing the very close interaction between PNS and peripheral digestive organs. Some remarks could be suggested to improve this manuscript:

  • Pancreatic ductal adenocarcinoma, which represented about 90/95% of pancreatic cancer, was derived from pre-cancerous lesions named PanIN (pancreatic intraepithelial neoplasia). Some reports have suggested that the nervous system could play a role in the proliferation of PanIN led to the development of cancer. This aspect could be evoked in the present review.
  • It has well known that chronic inflammation (Crohn’s disease and ulcerative colitis) of the colon represents a significative risk to develop colorectal cancer. It was shown that glial cells may have a role in inflammation and the maintenance of intestinal barrier. This aspect was not clearly evoked in the present manuscript.

Author Response

We are grateful for this reviewer’s very positive comments and valuable suggestions that, in our view, helped us to improve the manuscript.

“..Some reports have suggested that the nervous system could play a role in the proliferation of PanIN led to the development of cancer. This aspect could be evoked in the present review..”

Pancreatic pre-cancerous lesions (PanIN) are indeed the site where increased numbers of activated Schwann cells are observed. This association of glial cells with early lesions was well described in several publications by Demir and colleagues and linked to the nerve invasion process. We have referred to this work in the original manuscript and now we additionally stress this in the discussion of Schwann cell role in the exocrine pancreas.

“.. chronic inflammation (Crohn’s disease and ulcerative colitis) of the colon represents a significative risk to develop colorectal cancer. It was shown that glial cells may have a role in inflammation and the maintenance of intestinal barrier. This aspect was not clearly evoked in the present manuscript..”

This is a valid point because chronic inflammation of the intestines is a common condition that poses great health risks for the affected and is a field of active research in glial biology. We have included a paragraph about the involvement of ENS glia in Crohn’s disease and ulcerative colitis and discussed possible functions of Schwann cells in these conditions. Additionally, we mention the inflammatory disorders of the gut in the final section of the manuscript.

Reviewer 3 Report

In this review the authors produce a systematical review of the role of Schwann cell in the digestive system disorder. The review presents an objective status within the field with a major knowns and unknowns. In addition it suggests the paths for the future research.

Here is a list of minor issues that need fixing. In line 62, peculiar should be substituted by interesting.

PNI has been defined a perineurial invasion and as such labelled also in the figure 3. I learned perineural during my studies.

The term development (l. 65, l. 72) is used in a narrow embryonic development sense. I would add embryonic to clarify the issue.

There is a typo at the end of sentence in l. 69.

Author Response

We are grateful for the reviewer’s positive evaluation of our work.

We would like to thank the reviewer for pointing at the issues that needed fixing in the manuscript, including the spelling and correct word use. We have addressed those. Additionally, we performed the grammar and style editing of the text.

Point-by-point changes:

"In line 62, peculiar should be substituted by interesting."

Sentence was re-structured.

"PNI has been defined a perineurial invasion and as such labelled also in the figure 3. I learned perineural during my studies."

Changed to perineural in the text and the figure.

"The term development (l. 65, l. 72) is used in a narrow embryonic development sense. I would add embryonic to clarify the issue."

The term was specified by adding embryonic.

"There is a typo at the end of sentence in l. 69."

Fixed.

Reviewer 4 Report

This is a great and complete review with a number of nice illustrations. It really gets the reader into the field of the role of Schwann cells in digestive system. The only drawback includes insufficient referencing when it comes to the Schwann cell origin topic, the discussion around it and the multipotency of SCPs. Also, providing more in-depth review on the roles and diversity of enteric glia would be beneficial.  As far as I can see, the authors prefer to cite reviews instead of a number of original research articles, which is in principle understandable, although I would advise to include the references to the original articles as well. At least, I suggest to include the discussion of the following works: 

doi: 10.1007/s10286-017-0478-7. 

DOI: 10.1126/science.aah5454

doi: 10.1073/pnas.1710308114.

doi: 10.1016/j.stem.2021.12.003.

doi: 10.1038/s41586-021-04006-z. 

doi: 10.7554/eLife.56086.

doi: 10.1002/glia.23596.

doi: 10.1038/ni.3634.

doi: 10.1016/j.ydbio.2018.02.008.

Author Response

We are happy about this Reviewer’s overall very positive evaluation of our work. The Reviewer made very valuable suggestions for further discussion that we believe have considerably improved the manuscript.

Point-by-poin reply: 

“..drawback includes insufficient referencing when it comes to the Schwann cell origin topic, the discussion around it and the multipotency of SCPs..”

We included additional discussion on the developmental origins of Schwann cells and the multipotency of SCPs. We find the biology behind SCPs thrilling and feel that, despite major recent discoveries, there are many particularities that still need to be clarified. We believe that many interesting findings are still to come regarding the SCP/SC roles, particularly in the organs of the digestive system.

..Also, providing more in-depth review on the roles and diversity of enteric glia would be beneficial.  As far as I can see, the authors prefer to cite reviews instead of a number of original research articles, which is in principle understandable, although I would advise to include the references to the original articles as well..”

The diversity of phenotypes and functions of ENS glia is indeed very exciting. We have broadened the discussion on enteric glia and included more references to the original work that, in our opinion, is important. We have also added more discussion on the functions of enteric glia in the inflammatory disorders of the intestine.

“I suggest to include the discussion of the following works..”

We are grateful for pointing out some of this research for us, particularly the recent finding on the roles of GFAP+ and Plp1+ glia in the maintenance of the intestinal stem cell niche. The discussion on the nature of the pelvic and sacral innervation recently raised by Jean-Francois Brunet lab is also very interesting to follow. This has prompted us to make changes to Figure1, to include the pelvic ganglion. We were happy to refer to all of the suggested works in our manuscript.

Round 2

Reviewer 1 Report

1) General comments

Dr. Goluba and Dr. Parfejevs, et al. revised ‘Schwann cells in digestive system disorders’. This article is well presented. I read your responses for my questions. I understood your points of view in your study.